# Automatic 3D Building Model Generation from Airborne LiDAR Data and OpenStreetMap Using Procedural Modeling

Robert Župan *, Adam Vinković, Rexhep Nikçi and Bernarda Pinjatela

Faculty of Geodesy, University of Zagreb, 10 000 Zagreb, Croatia; adam.vinkovic@geof.unizg.hr (A.V.); rexhep.ini@t-online.de (R.N.); bernarda.pinjatela@geof.unizg.hr (B.P.)
* Correspondence: robert.zupan@geof.unizg.hr

**Abstract:** This research is primarily focused on utilizing available airborne LiDAR data and spatial data from the OpenStreetMap (OSM) database to generate 3D models of buildings for a large-scale urban area. The city center of Ljubljana, Slovenia, was selected for the study area due to data availability and diversity of building shapes, heights, and functions, which presented a challenge for the automated generation of 3D models. To extract building heights, a range of data sources were utilized, including OSM attribute data, as well as georeferenced and classified point clouds and a digital elevation model (DEM) obtained from openly available LiDAR survey data of the Slovenian Environment Agency. A digital surface model (DSM) and digital terrain model (DTM) were derived from the processed LiDAR data. Building outlines and attributes were extracted from OSM and processed using QGIS. Spatial coverage of OSM data for buildings in the study area is excellent, whereas only 18% have attributes describing external appearance of the building and 6% describing roof type. LASTools software (rapidlasso GmbH, Friedrichshafener Straße 1, 82205 Gilching, GERMANY) was used to derive and assign building heights from 3D coordinates of the segmented point clouds. Various software options for procedural modeling were compared and Blender was selected due to the ability to process OSM data, availability of documentation, and low computing requirements. Using procedural modeling, a 3D model with level of detail (LOD) 1 was created fully automated. After analyzing roof types, a 3D model with LOD2 was created fully automated for 87.64% of buildings. For the remaining buildings, a comparison of procedural roof modeling and manual roof editing was performed. Finally, a visual comparison between the resulting 3D model and Google Earth's model was performed. The main objective of this study is to demonstrate the efficient modeling process using open data and free software and resulting in an enhanced accuracy of the 3D building models compared to previous LOD2 iterations.

**Keywords:** LIDAR; procedural modeling; OSM; Blender; 3D model; buildings

## 1. Introduction

Procedural modeling, a sophisticated computer graphics technique, harnesses the power of algorithms, rules, and procedures to generate intricate objects and environments. Rather than painstakingly crafting each minute detail individually, this approach relies on the manipulation of parameters to establish rules and algorithms that give rise to the desired objects [1]. The versatility of procedural modeling extends across diverse domains, spanning the realms of film production, video games, architectural design, and urban planning [2]. One of its primary merits lies in its capacity to generate expansive and intricate virtual worlds that would otherwise be excessively arduous to construct through manual means. Moreover, the ability to swiftly adjust parameters and rules empowers creators to readily manipulate the resulting objects, facilitating experimentation and the exploration of multiple variations. Procedural modeling manifests itself in various applications, encompassing the generation of lifelike terrains for immersive video game experiences [2–4]; the creation of sprawling cities and animated settlements for cinematic productions [5]; the

precise modeling of buildings for architectural endeavors [6] and archaeological investigations [7,8]; and even the intricate synthesis of natural phenomena, such as trees, mountains, and rivers, each with its unique forms and characteristics [3,9]. In the past two decades, the advent and refinement of 3D city models have profoundly impacted urban development processes, transportation planning, environmental stewardship, and tourism, thanks to the vast range of possibilities they afford.

The environment in which we find ourselves is increasingly complex in its economic, infrastructural, and social sense; and the abundance of information we collect about it needs to be modeled, stored, and distributed for large areas, such as entire city areas. For this purpose, 3D models of cities appear as an information repository that can be used for numerous purposes, such as:

- urban (built-up) analyses [10];
- 3D urban morphology change [11];
- management of city districts [12,13];
- development of tourism [14];
- traffic [15];
- cadastre [16];
- cartography and mapping [17];
- architecture and urban planning [18–20];
- environmental quality [21];
- infrastructure planning [22];
- heating demand prediction [23];
- solar potential analyses [24], etc.

For a comprehensive overview of use cases of 3D city models, see Biljecki et al. [25].

With the widespread availability of data produced by LiDAR (light detection and ranging) sensors, there has been interest in automatic construction of models of urban areas [19,26,27]. LiDAR data marked the beginning of rapid collection and disposal of spatial information for large areas. LiDAR technology uses laser light to measure distances and create detailed 3D models of objects and environments. In the context of building modeling, LiDAR data can be used to create highly accurate and detailed 3D models of buildings, which can be used for a variety of purposes, including building extraction [28], building reconstruction [29,30], change detection [31], and urban analyses [32].

With the increase in resolution of point clouds that can be collected by aerial photography using LiDAR technology, the potential for automatic generation of building models with a higher level of detail (LOD) has emerged. For some applications, simple forms without details are sufficient. But some applications require a higher LOD; e.g., when analyzing the solar potential of the surfaces of roofs, is it necessary to have a detailed model of the roof that includes chimneys, windows, and similar structures? Thanks to advances in photogrammetry and remote sensing, point clouds are becoming widely available and bring a potential that needs to be explored in the sphere of urban modeling [33]. Processing and visualization of data derived from aerial photographs, however, remains a problem and brings great challenges. Automating and accelerating urban model generation from point cloud data, with the aim of respecting smaller details on buildings that until now, could only be modeled by user intervention, are the main goals of this research.

LiDAR technology has been widely researched and applied for 3D modeling in various fields and in combination with different data sources. Vosselman and Maas [34] present a comprehensive review of LiDAR-based 3D modeling of buildings. They discuss various methodologies, data acquisition techniques, and data processing algorithms for generating accurate and detailed 3D models of buildings using LiDAR data. Haala and Anders [35] explored the three-dimensional reconstruction of buildings from aerial images, digital surface models, and existing 2D building information. Chen et al. [36] investigate the fusion of LiDAR and optical imagery for building modeling. In their research, they propose a novel SMS (split–merge–shape) method for building detection and building reconstruction. Dorninger and Pfeifer [37] developed an automated approach for 3D building extraction

from airborne LiDAR point clouds based on a 3D segmentation algorithm that detects planar faces in a point cloud. Mathews et al. [38] use LiDAR data in combination with satellite scatterometer (radar) data to estimate 3D urban built-up volume. Several other researchers have tried to automate 3D building modeling with airborne LiDAR data [39,40], as well as analyze their quality [41] and accuracy [42]. A recent study by Barranquero et al. [30] uses a convolutional neural network to analyze LiDAR data and supplement it with OSM data to automatically reconstruct 3D urban environments.

The automatic generation of very detailed 3D models based on real spatial data has been the subject of numerous scientific works over the past twenty years. At the beginning of the twenty-first century, researchers emphasized mainly the processing of photogrammetric images by classifying and segmenting surfaces with algorithms such as RANSAC (random sample consensus). The RANSAC algorithm detects regular geometric shapes such as a line or circle from a given 2D image or a plane in 3D space [43]. Tarsha-Kurdi et al. [43] used this algorithm for automatic detection of building roofs from LiDAR data. While the RANSAC algorithm can be found in the form of open code, they needed to adapt and extend it to search for surfaces that best match the roof geometries, which turned out to be extremely complex.

In the work of Rychard and Borkowski [44], an automatic procedure was developed that discerns and semantically interprets the structures that make up buildings by constructing the surface on which the observed point is located. It starts from the original point, which is part of the area to be built, and candidate points are added to it if they meet the criteria visible in the algorithm. If the candidate points are part of the same plane, they are counted flat, and the analysis moves to the next point. An important part of this scientific work is the automation of roof structure recognition. Parts of buildings developed by the surface construction algorithm are assigned by means of topological graphs to the corresponding, predefined structures from the semantic repository. In other words, the point cloud segmentation products are assigned to the corresponding 3D models (Figure 3.2 in [44]).

In his thesis, Wichmann [45] uses a sub-surface growing approach by extending and modifying the 3D Hough transformation to reconstruct a 3D model from a LIDAR point cloud. He defines sub-surfaces as surfaces that extend below common additional roof contents such as antennas and chimneys, and which are important for properly defining the roof geometry, as well as for removing additional roof contents during segmentation. With this approach, holes in the data are "patched" by creating virtual points.

An interesting example of the application of procedural techniques in 3D modeling of buildings is the research of Wu et al. [46], which investigates the inverse procedural modeling of building facades using split grammar, with the aim of finding procedural descriptions for the observed model. Facade images are used to break down basic elementary forms and regularities of buildings to generate a set of modeling rules (Figure 6 in [46]) and apply them to experimental models.

An example of the use of procedural modeling in the practice of spatial planning is the project of creating a digital twin city for the Kalasatama district in Helsinki, Finland [47]. The goal was to create a city model with semantic data in accordance with CityGML, an open standard for 3D modeling, registration, and distribution of spatial data in which an object is associated with geometry, semantics, topology, and display mode [47]. During the creation of the model, all parts of the area were first photogrammetrically recorded with a spatial resolution of 6.5 cm. The area was then broken down into 250 m $\times$ 250 m squares, easily recognizable and isolated control points measured by hand on the physical surface of the Earth were added. Aerial triangulation of all the squares was performed. By connecting them into a whole through common points, the network that makes up the model was created. Parameters were optimized so that the model corresponded to real objects, and the quality was checked by visual inspection of coordinates for each object. Therefore, user intervention was required during the otherwise automated process. While the study primarily showcases a substantial increase in the utilization of automated techniques for

3D modeling of buildings, we have endeavored to outline all the steps involved in the manual manipulation process. The Discussion section elucidates the key advantages of embracing automation within procedural modeling, along with supplementary resources that integrate and automate the aforementioned steps through programming, providing a comprehensive framework.

## 2. Materials and Methods

### 2.1. Study Area

As the study area, we chose the city center of Ljubljana since LiDAR data is freely available for the complete territory of Slovenia. Ljubljana is the largest city and capital, with a population of around 300,000 and covering an area of 164 km² (https://en.wikipedia.org/wiki/Ljubljana, accessed on 15 May 2023). For our research, we selected the area in the city center, which is bounded on the north side by Tivolska and Masaryk road; on the east side by Resljeva road; on the south side by Cankarjeva road, Čopova street, Prešern square, and Petkovškovo nabrežje; and on the west side by Bleiweiseova road (Figure 1). The mentioned area was chosen due to the variability of the shapes, heights, and functions of the buildings, which represents a challenge when trying to automatically generate a 3D model.

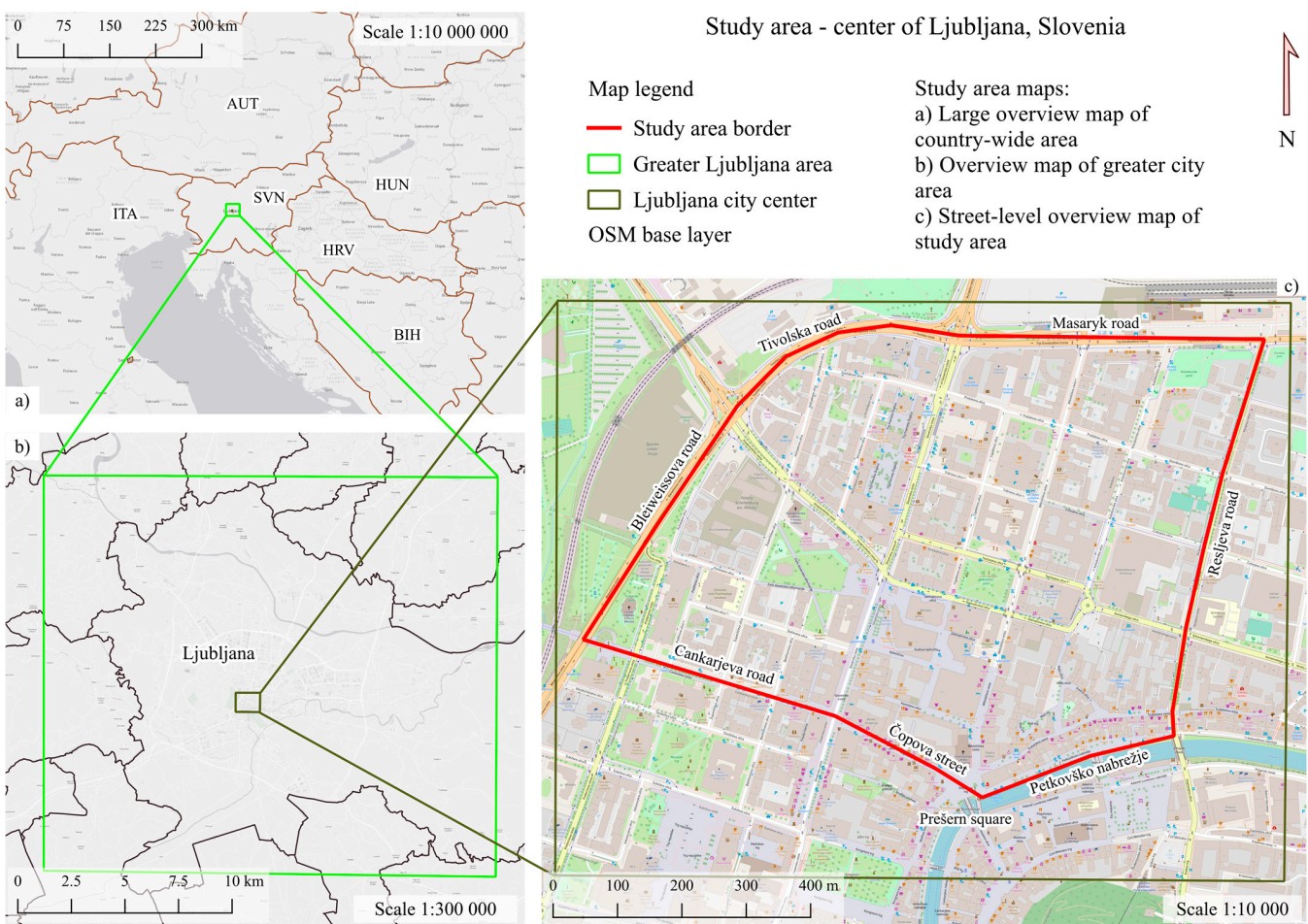

**Figure 1.** The border of the analyzed area in the center of Ljubljana, Slovenia is shown as a red line (© OpenStreetMap contributors).

### 2.2. Data Processing

LiDAR (light detection and ranging) data from the website of the Ministry of the Environment of the Republic of Slovenia [32] was downloaded with a spatial resolution of 5 pt/m². Using a web-map viewer on the website, it is necessary to select the area, data

types, and formats of the point cloud data collected by aerial photography. The data are divided into squares (tiles) with an area of 1 km$^2$, and can be downloaded as 3 different data types:

- OTR—(Oblak Točaka Reljefa) georeferenced relief point cloud containing only points classified at the ground (the storage format is zLAS);
- GKOT—(Georeferencirani i Klasificirani Oblak Točaka) georeferenced and classified point cloud, which includes points from the ground, buildings, and three different types of vegetation (the storage format is zLAS);
- DEM (digital elevation model (DEM), which is an interpolation of the relief based on OTR points), stored in a regular grid of 1 m × 1 m in the form of an ASCII file.

Note that zLAS is a compressed form of the LAS (LASer) format. The reason for the compression lies in the size of the data, which, despite the lower resolution, cannot be stored on the website of the Slovenian Ministry in its initial form. The data download procedure includes selecting the tile for the required area and the download option in OTR, GKOT, or DEM format. To cover our research area, we downloaded tiles TM_461_101, TM_461_102, TM_462_101 and TM_462_102. For these four tiles, a georeferenced point cloud, with classification of all points, and a DEM were downloaded. CloudCompare, a free open-source software for processing 3D point clouds (https://www.danielgm.net/cc/, accessed on 14 May 2023), was used to process the point cloud. Some of the processing options include reconstructing surfaces from point clouds, calculating volumes, and estimating geometric features of objects. The fragmented clips are separately uploaded to CloudCompare for the classified point cloud and the DEM, and then joined into a whole and roughly cut to the area to be analyzed. The output data is a classified cloud with 7,627,691 points; i.e., a digital surface model (DSM) and a digital terrain model (DTM) with 721,807 points, seen in Figure 2.

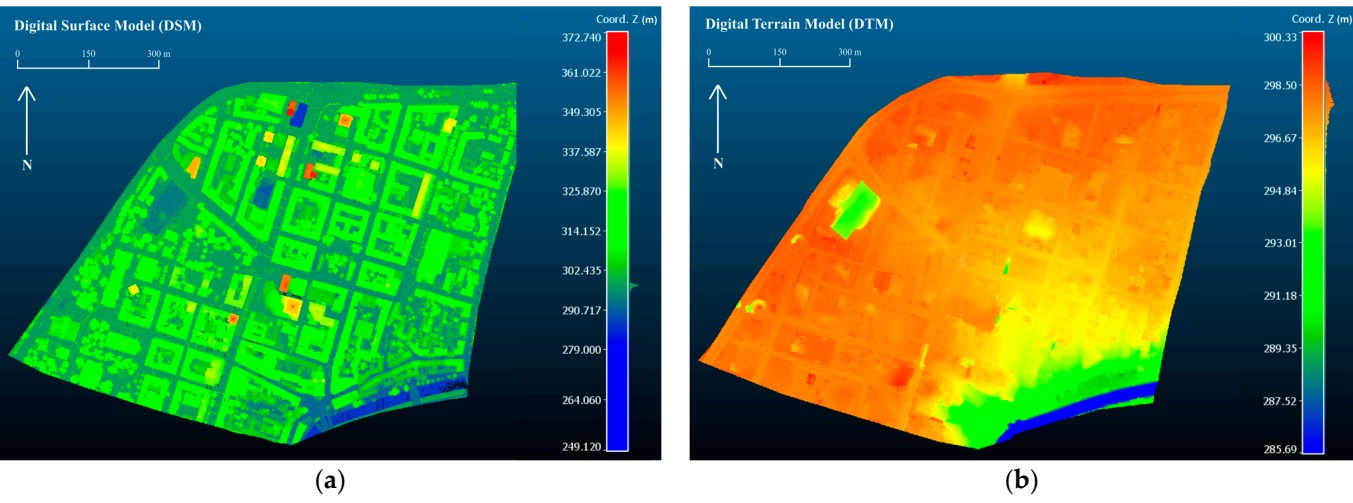

**Figure 2.** Output data (elevation) of the point cloud processing in CloudCompare: (**a**) DSM; (**b**) DTM.

To mask the preprocessed LIDAR data only to buildings in our study area, we had to download a vector layer with the outlines of the buildings. The idea was to add the calculated building height to the masked point cloud. This part of the data processing was performed in QGIS (quantum geographic information system)—a free and open-source GIS software that offers the benefit of seamlessly integrating various types of data (https://qgis.org/, accessed on 14 May 2023). To download a vector layer with outlines of the buildings in our study area, we used OSM Downloader plug-in (https://plugins.qgis.org/plugins/OSMDownloader/, accessed on 14 May 2023). Using this plug-in inside QGIS, a user can browse OSM data by area and download shapefiles by geometry type (point, line, multiline, multipolygon, etc.). Floor plans of buildings are located, together with floor plans of meadows, public areas, and parking lots, in the

multipolygon layer. After loading and for the purpose of realistic objects, one needs to edit the layer in a manner that allows for the extraction of building objects from the attribute table through multiple selection. These selected objects should then be saved as a separate layer. To ensure accurate georeferencing, it is essential to set the Slovene national grid as the projection, using the EPSG (European Petroleum Survey Group) code 3794.

The result of OSM data editing are 453 objects representing buildings. The attribute table contains a total of 25 columns with data about name and type of the object, geological features, area, land use, as well as fields with the names 'craft', 'leisure', 'man -made', 'military', 'place', 'shop', 'other tags', and other object descriptions not used in this research (Figure 3).

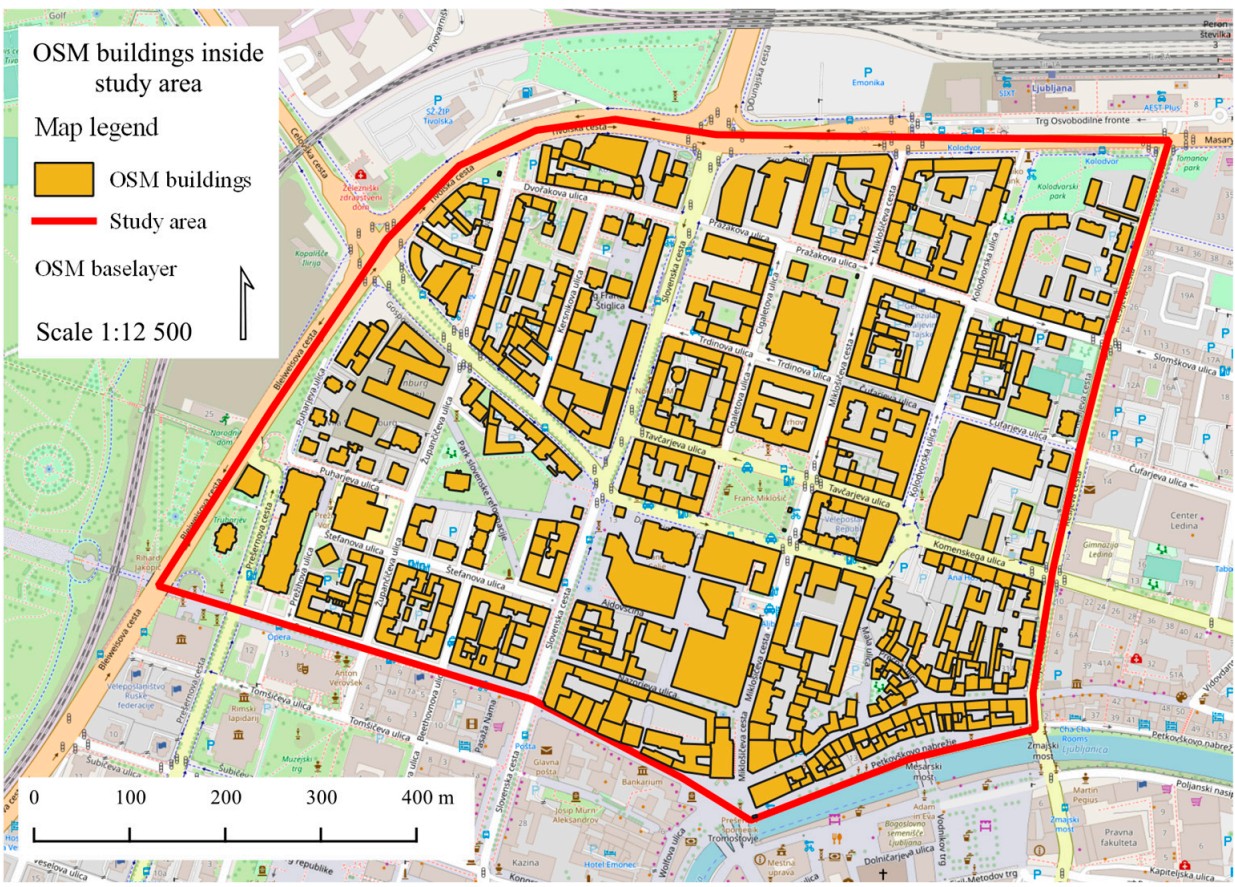

**Figure 3.** Shapefile layer with floor plans of buildings after filtering OSM data (© OpenStreetMap contributors).

The next important step was to assign the height to each building as a value in the attribute field. For this we used LASTools, a powerful collection of tools for processing LIDAR data (https://rapidlasso.com/lastools/, accessed on 14 May 2023). First, the point clouds representing the DEM and the classified point cloud from the las format were converted into a raster file with pixel values that correspond to Z coordinates from the point cloud. For this purpose, the LAStools plugin (https://plugins.qgis.org/plugins/LAStools/, accessed on 14 May 2023) was installed inside QGIS, after which it was possible to access the tool via the Processing toolbox. Using Lasview within QGIS we obtained a DSM, that is a display of 3D coordinates needed for further point segmentation (Figure 4).

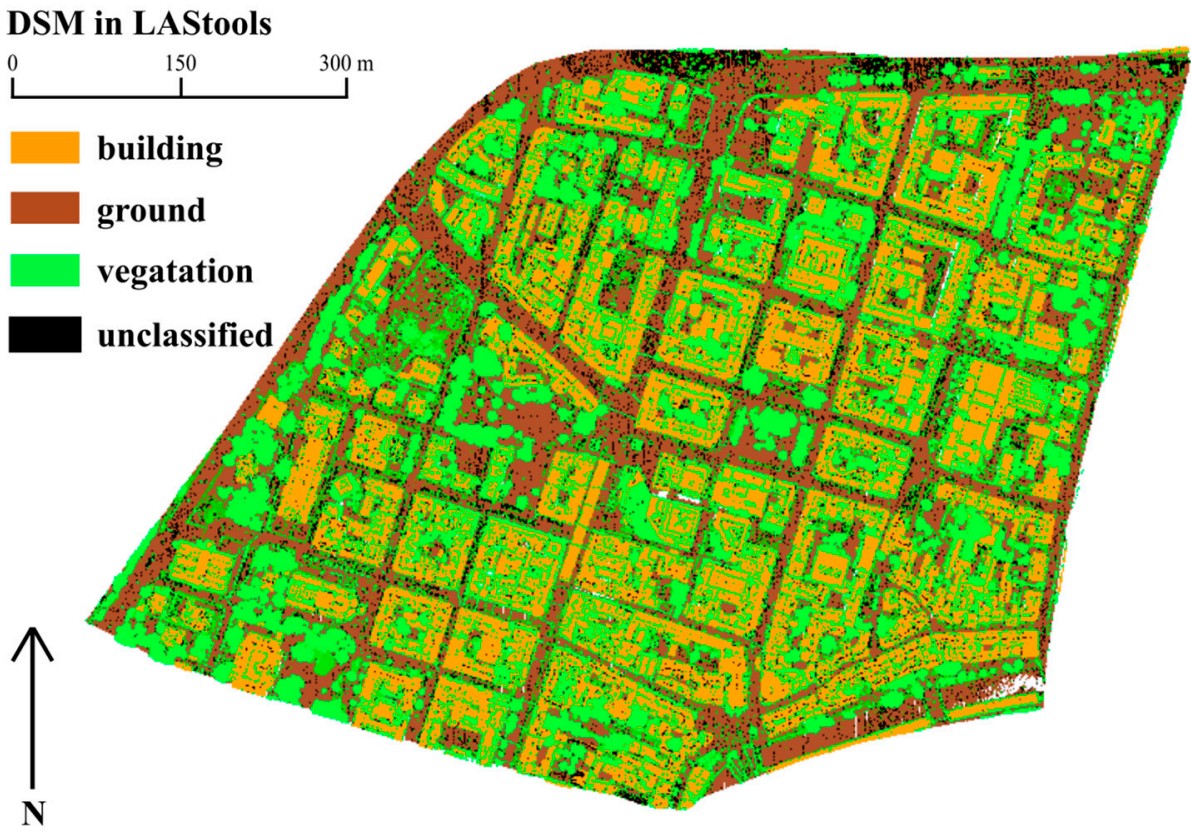

**Figure 4.** Classified DSM in LAStools.

In the next step, we utilized Lassplit. As LAStools is licensed software, the maximum point cloud size that can be processed is limited to 1,500,000 points. Therefore, it was necessary to divide the classified point cloud into six parts to adhere to this limitation. The resulting clips were saved in a shared directory to expedite the subsequent conversion to raster format.

After dividing the classified point cloud, we applied Las2demPro to the point clouds in a version that processes the entire directory at once. The input parameter is a directory containing six parts of the point cloud in *las* format, and the attribute selected for display in the raster file was the elevation. For the digital terrain model, a version that processes a single file was used, since fragmentation of the point cloud was not necessary here. The result is an image related to each of the six classified point cloud parts and one image for the DEM. We then loaded the raster data into QGIS and assigned the appropriate D96/TM projection with EPSG code 3794. After creating a continuous image for all classified point cloud parts using the Build Virtual Raster method, we compared the elevations in the areas where buildings are located. The range of pixel values for classified points is from 285 to 353 m in height and for the DEM, from 291 to 300. Brighter pixels correspond to higher values for heights. By subtracting these two images, we obtained the building heights; i.e., the height values of pixels from the classified point cloud were simply subtracted for the height values found in the DEM. The resulting image is shown in Figure 5.

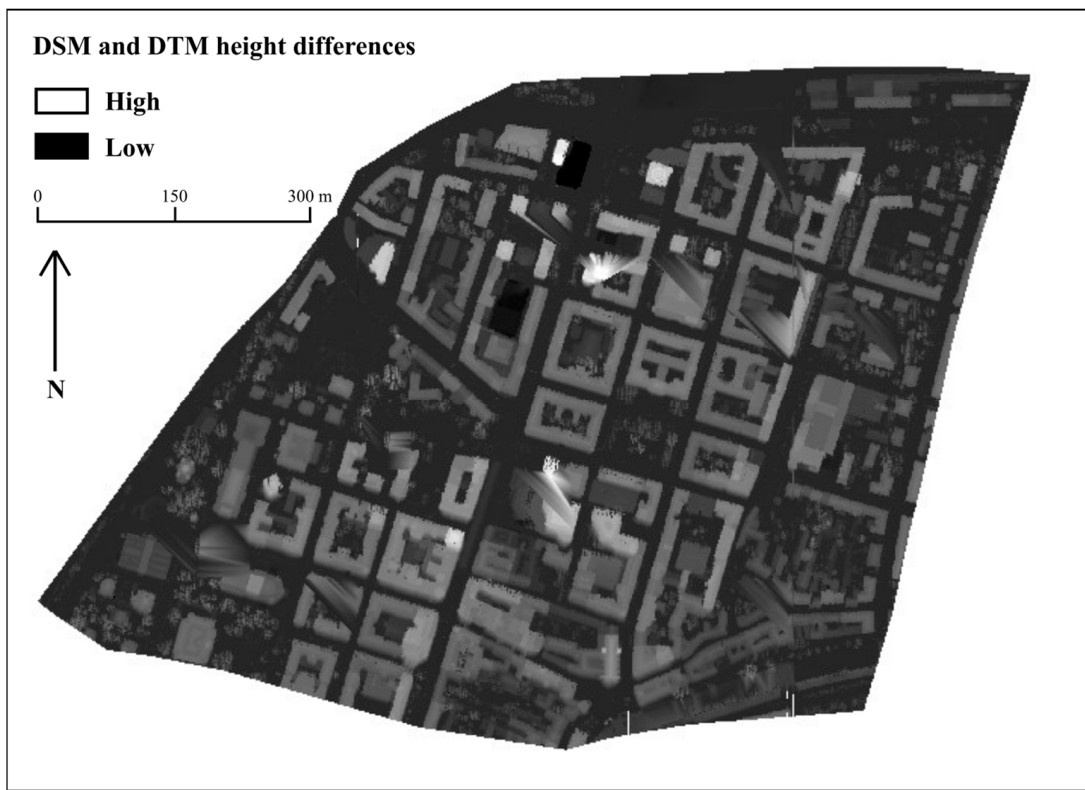

**Figure 5.** Differences by height between DSM and DTM.

Next, we compared the elevation differences to the building boundaries to assign heights only to the areas of the buildings using the Zonal statistics tool. The raster layer with elevation differences and the vector layer with zones (i.e., the layer containing the buildings) were chosen for analysis. Moreover, it is possible to select and filter the statistical parameters before further calculations. In this case, we selected the minimum, maximum, and mean values to distinguish the actual roof height from the heights of antennas and other structures typically present on building roofs. These values are then displayed as results in the attribute table of the buildings.

In addition to building heights, an important item in automated building modeling is the roof shape. Considering that only 25 of the 453 buildings had the roof type attribute, it was necessary to add the values of the attribute field for all other buildings included in the analyzed area. Google Earth was used for a detailed inspection of roofs and entry of values for buildings on which the type of roof was not marked.

### 2.3. Comparison of Software for Procedural Modeling

One of the research goals was to automatically model buildings based on LiDAR data in the simplest possible way and with readily available software. When considering software choice, we set several main conditions that must be met. The software of choice must be able to: generate complete buildings in 3D; import geographic and attribute data in a GIS format to utilize building footprints (outlines) and building height information; and approximately model roof geometries depending on their form. Different software for automatic creation of 3D models were compared by characteristics, such as ease of use, price, availability of learning materials, and the ability to process 3D point clouds as well as OpenStreetMap (OSM) data. After narrowing down the selection, we decided to compare: Houdini, CityEngine, Unity, Maya, Blender, Geopipe, Omniverse, Mapbox, and Cesium.

The most famous computer programs for procedural modeling are certainly Houdini and CityEngine. Houdini enables the creation of 3D models and animations; the creation of lighting and particles; the simulation of phenomena such as clouds, smoke, and fire;

and numerous upgrades depending on the needs of the user to expand functions (https://www.sidefx.com/products/houdini/, accessed on 15 May 2023). The operators on which Houdini is based are organized into nodes and allow the user to create complex geometry in a small number of steps, where the development of 3D objects and scenes does not have to be linear—by changing just one of the parameters, the user can create a whole series of new objects in the scene. Houdini supports the manipulation of point clouds in ply format. When considering the price of the software, different options are offered depending on the commerciality of use, and one of them is Houdini Apprentice—a free and limited version for students and recreational users who want to use the software for the purpose of learning, research, and for non-commercial creation of 3D models and animations. By looking at the possibilities offered by Houdini Apprentice, it was determined that the version is sufficient for creating 3D models based on LIDAR data, and by using plugins from GitHub, one can load shapefile layers containing OSM data. However, due to the complex interface and the very time-consuming process of mastering Houdini functions, this software was ultimately not chosen for the creation of the 3D model.

During software comparison, ArcGIS CityEngine by Esri (https://www.esri.com/en-us/arcgis/products/arcgis-cityengine/, accessed on 15 May 2023) has to be considered as the first and most suitable choice. CityEngine is a software application used for 3D modeling and urban planning. Advantages include powerful procedural modeling capabilities, integration with GIS data, and real-time visualization. The biggest advantage of CityEngine is the built-in capability to recognize roof types from point clouds. However, while CityEngine excels at generating large-scale city models, it may not be the ideal tool for detailed modeling of individual buildings or complex architectural features. Also, there are very few learning materials available that focus on procedural building modeling, and the automatic roof recognition option cannot be fully exploited due to the low resolution of the processed LIDAR data. Another disadvantage is the cost of ArcGIS CityEngine Pro version, which comes at a price of US$100 per year for individual users.

Unity is a platform for creating video games in 2D and 3D environments (https://unity.com/, accessed on 15 May 2023). In the context of this work, the most interesting is the Unity extension CityGen3D, which contains tools for simple and automatic creation of scenes and cities based on OSM data. No programming knowledge is required to create the model, and the creation interface is very simple. In addition to OSM data, users can load digital terrain models and buildings textures. Additional contents that enter part of the city inventory, such as lighting, sidewalks, and benches, are added in a few simple steps; and by geometry deformations, it is possible to add a third dimension to objects such as railway tracks and sidewalks to enhance visualization. The disadvantage of this add-on is the fact that the outlines of buildings in Unity are imported directly from OSM data, and it is not possible to edit their attribute tables and thereby bring their final appearance closer to the actual situation on the ground. In addition, the price of this add-on is $125 (in 2023).

Autodesk's Maya (https://www.autodesk.com/products/maya/overview, accessed on 15 May 2023) was considered for 3D modeling. Specialized in 3D animation, this program meets the criterion of being able to load shapefiles and LIDAR point clouds, and the interface is intuitive to use. The disadvantage of this option is that there is not enough material available to master 3D modeling for a large area with many objects and with an emphasis on the LIDAR point cloud.

Blender is a free and open-source software for 3D modeling and animation, simulation, and rendering (https://www.blender.org/, accessed on 15 May 2023). Finally, it was selected based on several factors: availability, user-friendliness, abundance of online documentation and learning resources, capability to handle OSM data, and minimal hardware requirements. For the creation of 3D models of buildings, we used an interface for advanced procedural modification of geometry using a nodes system called Geometry nodes. This interface allows animators to perform procedural modeling that previously required more complex commercial programs such as Houdini. It appears for the first time in Blender in

version 2.92, released on 25 February 2021 (Blender Institute B.V., Buikslotermeerplein 161, 1025 ET Amsterdam, The Netherlands).

At the heart of Geopipe's technology is its advanced machine learning algorithms and data processing techniques. By leveraging vast amounts of geospatial data, such as satellite imagery, LiDAR scans, and other sources, Geopipe can reconstruct real-world locations in stunning detail. This process involves capturing the intricate geometry, textures, and semantic information of the environment, ensuring a true-to-life representation.

One of the key advantages of Geopipe's approach is its ability to rapidly generate 3D models at scale. Traditional methods for creating virtual environments often require manual labor and expertise, resulting in significant time and cost investments. Geopipe's automated pipeline streamlines this process, enabling efficient generation of virtual worlds for various applications. There is no information on the cost of that technology (https://www.geopipe.ai/about, accessed on 25 July 2023).

Omniverse, developed by NVIDIA, is a groundbreaking platform that aims to revolutionize collaboration and simulation in various industries. Launched in 2020, Omniverse enables real-time, multi-user, and cross-domain collaboration, allowing teams to work together seamlessly in a shared virtual environment. At the core of Omniverse's capabilities is its powerful real-time 3D simulation engine. This engine, known as the NVIDIA RTX renderer, harnesses the immense computing power of NVIDIA GPUs to deliver stunning visual fidelity and realistic physics simulations. It enables users to create and interact with virtual environments that closely resemble the real world, enhancing the design and decision-making processes. One of the key features of Omniverse is its ability to integrate and synchronize diverse software tools and workflows. It serves as a common platform that bridges the gap between different applications, allowing professionals from various disciplines to collaborate effectively. With Omniverse, architects, designers, engineers, and artists can work together simultaneously, sharing their designs, making modifications in real time, and seeing the impact of changes instantly (https://www.pny.com/en-eu/professional/software/nvidia-omniverse-prod, accessed on 25 July 2023).

Mapbox has emerged as a leading provider of location data and mapping services, empowering businesses and developers to create highly customizable and interactive mapping experiences. With its powerful technology, extensive product offerings, and commitment to collaboration, Mapbox is driving innovation in the field of location-based services. As the demand for location data and mapping experiences continues to grow, Mapbox remains at the forefront, enabling businesses to harness the power of location intelligence, and enhance their applications and services (https://www.mapbox.com/, accessed on 25 July 2023).

Cesium is widely adopted across various industries and applications. In urban planning and architecture, Cesium enables the creation of interactive 3D models of cities, allowing stakeholders to visualize and evaluate proposed developments in their real-world context. It also finds applications in defense and intelligence, where it supports mission planning, terrain analysis, and situational awareness (https://cesium.com/, accessed on 25 July 2023).

## 3. Procedural Modeling in Blender

To link the processed data with Blender, the BlenderGIS add-on was used. BlenderGIS add-on is the most important component that allows simplicity of procedural modeling inside Blender without the need for programming knowledge. The user can download add-on for free from Github (https://github.com/domlysz/BlenderGIS, accessed on 15 May 2023) and is widely used to connect Blender with spatial (geographic) data. Some functionalities provided by BlenderGIS include operators such as Basemaps, Get OSM and Get SRTM, which allow direct download of an OGC web map, OSM data, or SRTM data (DEM's from NASA). Furthermore, BlenderGIS facilitates import and export of shapefile

layers, georeferenced images, and OSM data. Also, coordinate projections are available, which is important for object placement on the terrain, i.e., on real world coordinates.

When importing a shapefile file using BlenderGIS add-on, we had to select the attribute field that contains height data. In our case, this is the building height calculated in the earlier steps. The offset option in relation to the value of the selected attribute field was not used in this step, considering that the objects will (later) be placed on the reference surface. Finally, we split objects into individual units or entities, and selected the appropriate coordinate system. The result is an imported 3D model with heights and floor plans corresponding to buildings in the real world (Figure 6).

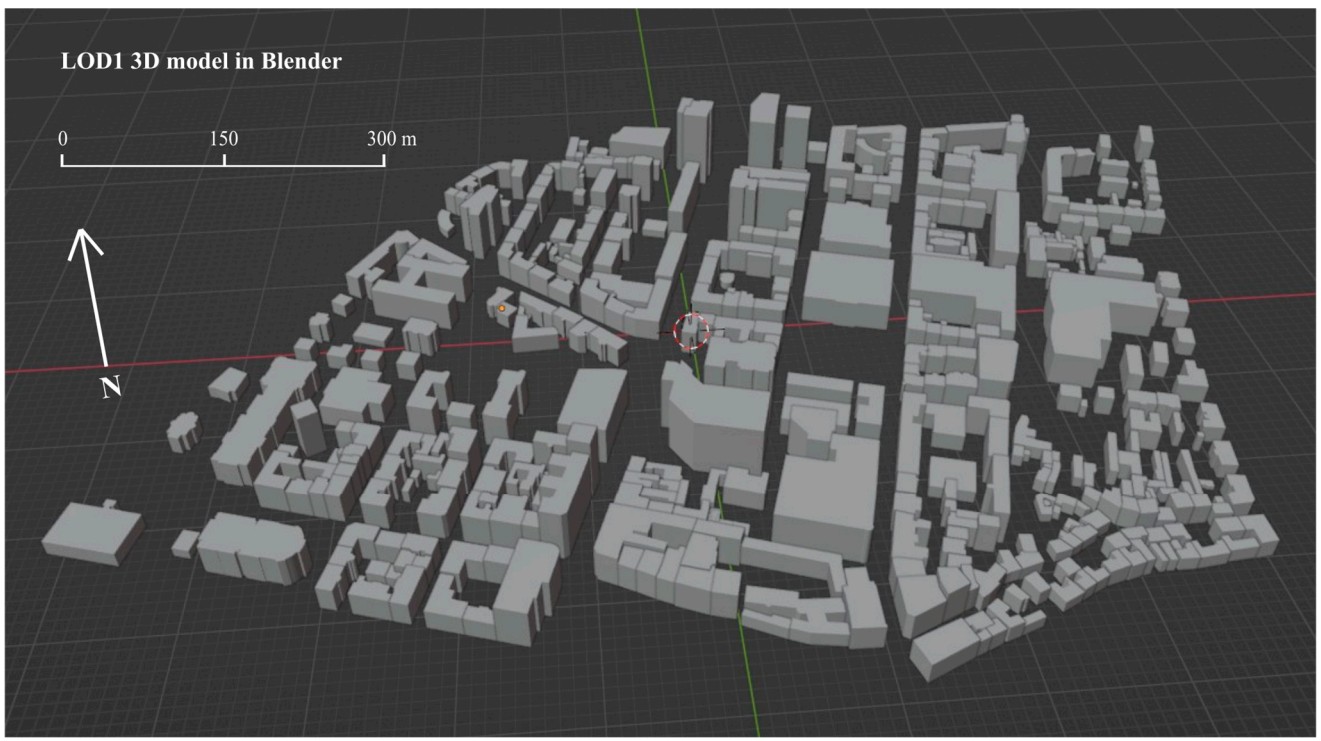

**Figure 6.** 3D building model imported in Blender.

The next step is to model the roofs using a procedural technique (Figure 7). Of the numerous existing forms of roofs that appear in architecture, four types of roofs were recorded in the analyzed area (Figure 7): flat, tented or pitched, gable. Tented and pitched roofs are classified in the same category, given that the geometric rules required for their generation are the same, and the outcome depends only on the floor plan—tent roofs occur in squared and pitched roofs in rectangular floor plans. Flat roofs are the most common and account for 286 buildings in the observed area. Besides them, there are 150 tented or pitched roofs, and only 17 gable roofs.

Due to simple roof geometry of flat roofs, to model them it was necessary to: (a) select all 286 buildings with this type of roof; (b) separate roofs from walls using the Edit mode; (c) save them in separate layers; and (d) open the *Geometry nodes* interface and model the walls with a height of 1.3 m by hollowing out the roof towards the building foundation by 1.3 m. The walls of all buildings within this category are 20 cm thick for a more realistic representation of the facade (Figure 8).

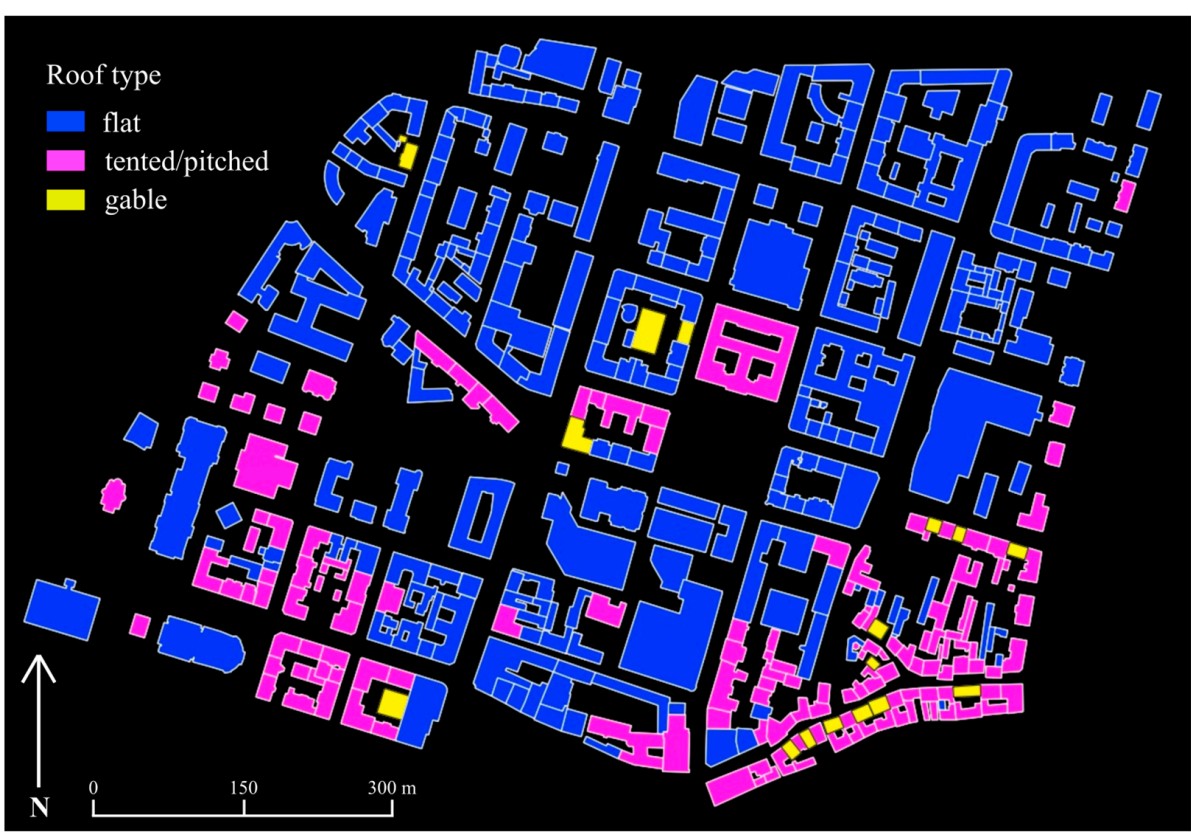

**Figure 7.** Roof types in the analyzed area.

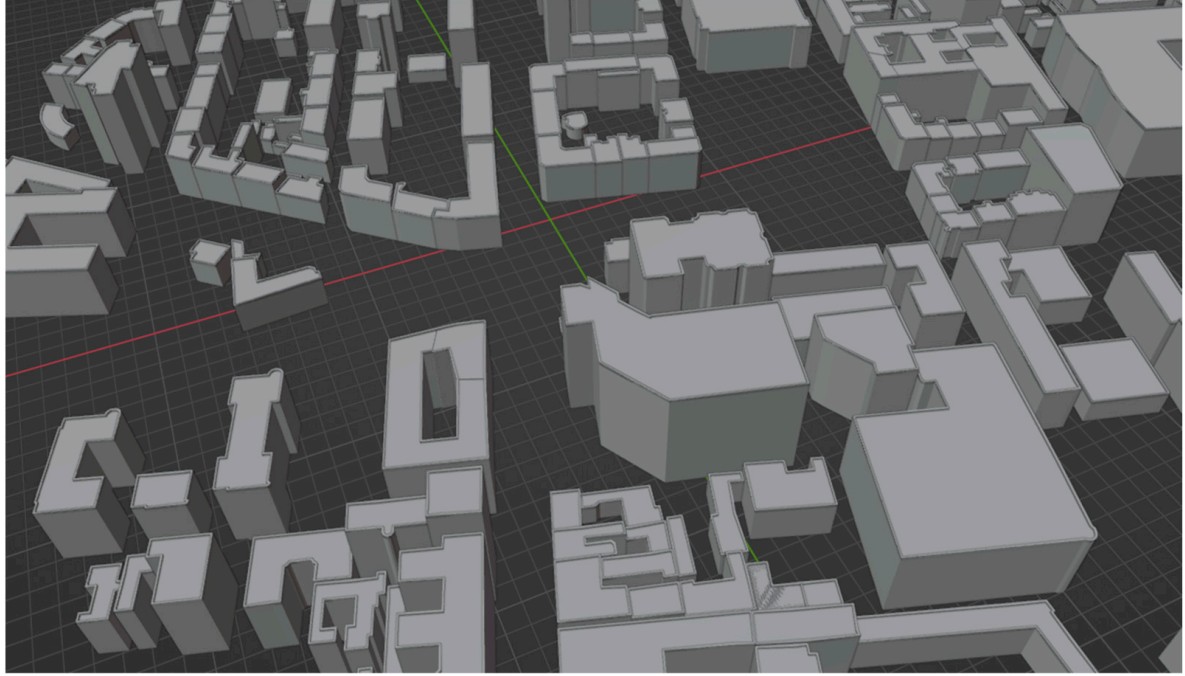

**Figure 8.** The result of mathematical rules applied to flat roofs.

With gabled roofs, we divided the surface into two parts, selecting hubs that contain two surfaces and raise them by 3 m. This is conducted by combining several modifiers in the Geometry nodes interface (Figure 9), the result of which is a simple model with a fixed height of the roof ridge (Figure 10). For the model to correspond more closely to the real object, the roof was additionally raised in accordance with the maximum and average height values in the attribute table for each of the 17 examples.

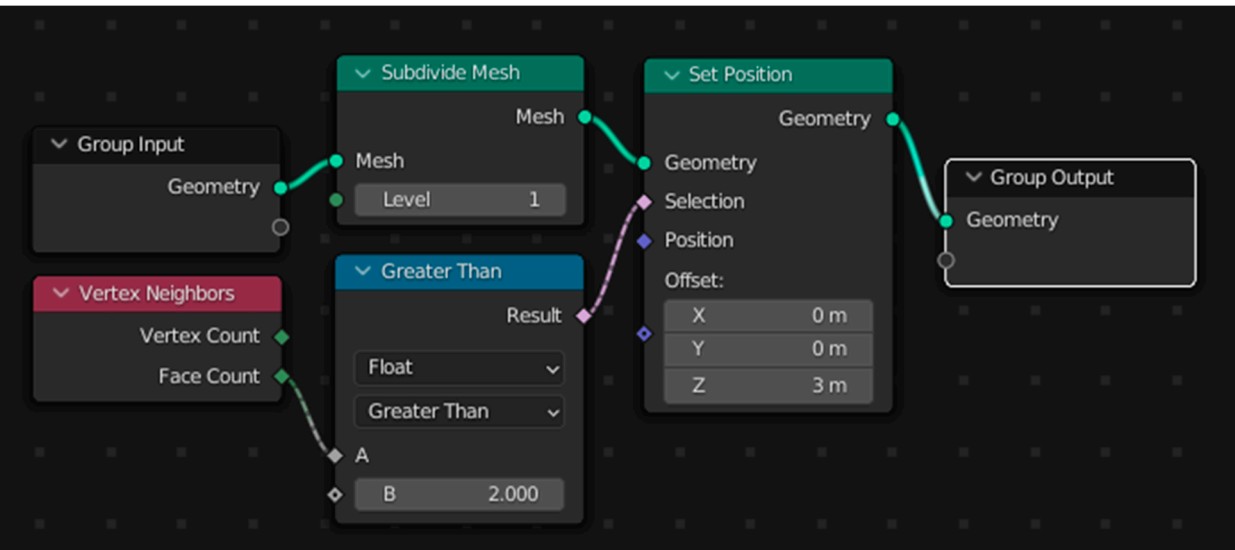

**Figure 9.** Settings within the geometry nodes interface for gable roofs.

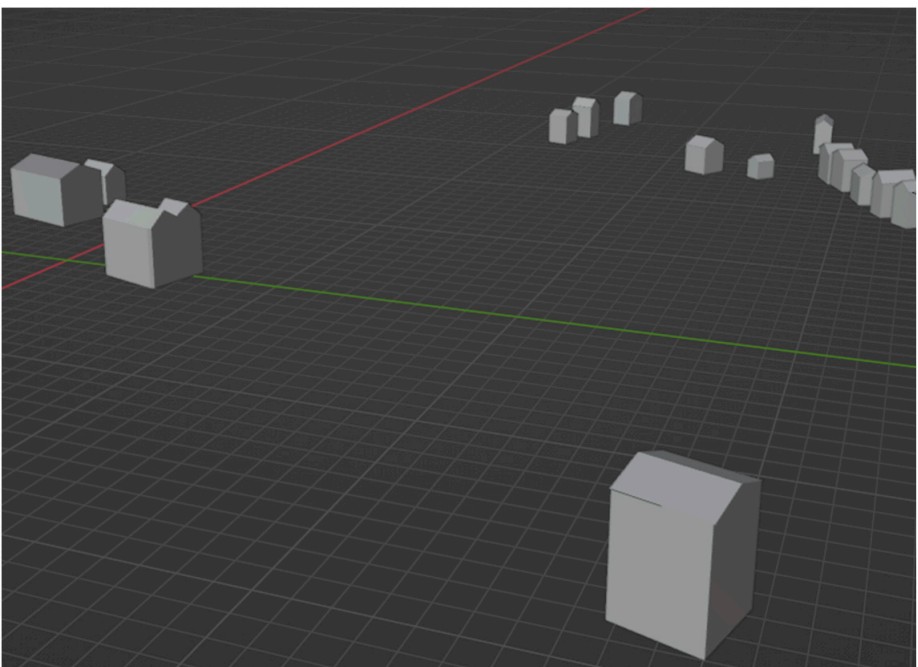

**Figure 10.** The result of mathematical rules applied to gable roofs.

For pitched and pitched roofs, a surface triangulation modifier was used within the Geometry nodes interface (Figure 11).

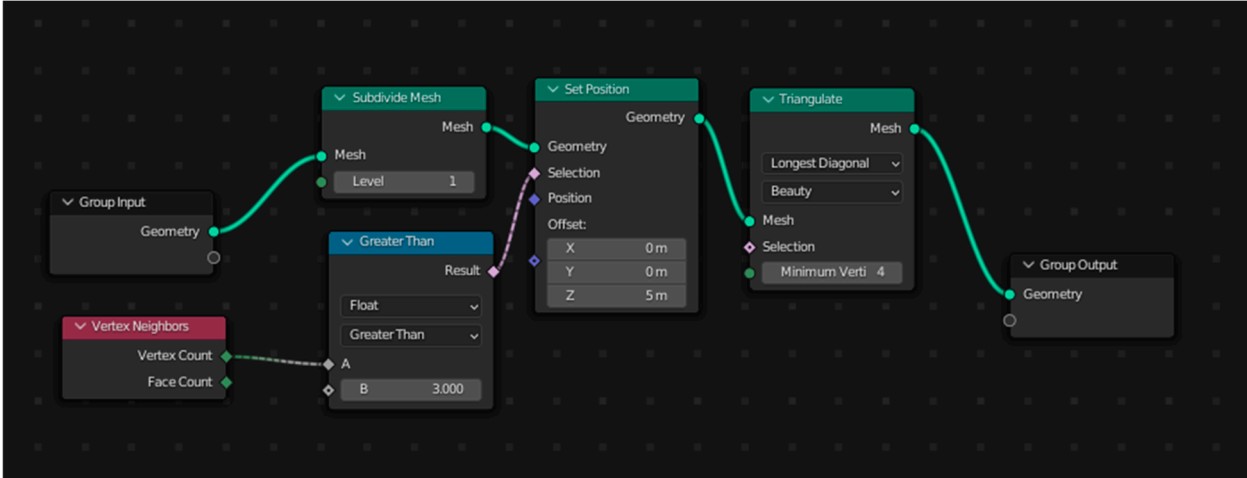

**Figure 11.** Settings within the geometry nodes interface for tented and pitched roofs.

The triangulation of the surfaces and the elevation of the edges that correspond to the set criteria proved to be a simple and satisfactory solution for buildings whose contours did not contain protruding and elongated parts. In buildings with protruding parts of the facade, triangulation led to excessive roof edges (Figure 12a); it was necessary to manually edit the results (Figure 12b).

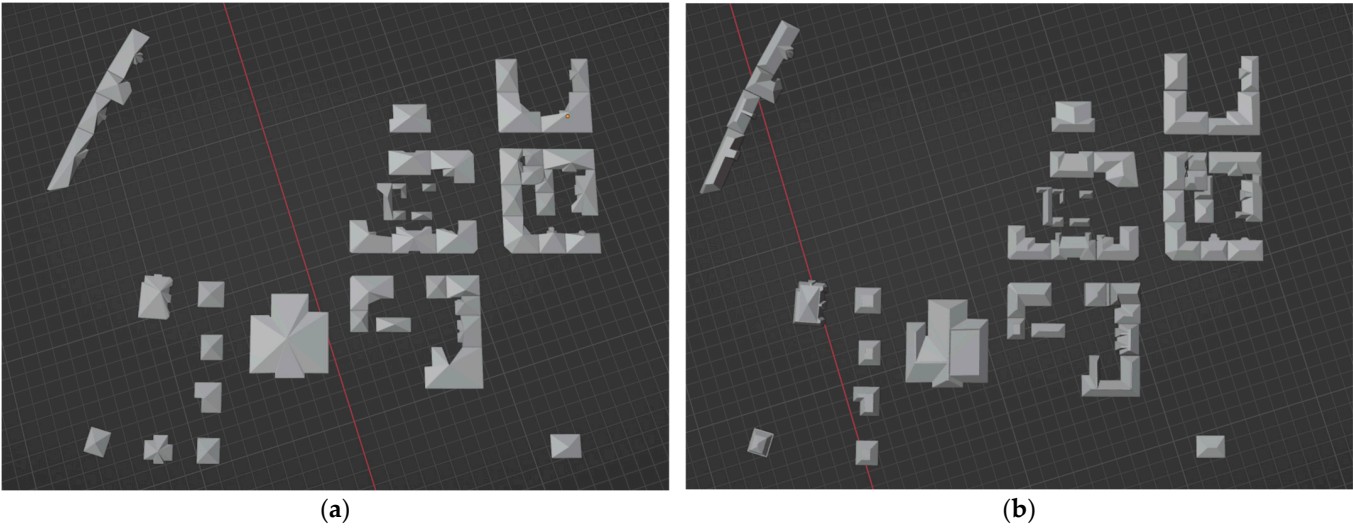

(**a**)                                                                                     (**b**)

**Figure 12.** Visualization of tented and pitched roofs in Blender after: (**a**) mathematical rules were applied; (**b**) manual editing.

The fourth group of buildings by type was counted as buildings with gabled roofs, but their geometry could not be produced automatically. In this group, there are sacred buildings with complex facades, floor plans, and roofs; so, it was necessary to model them manually using the functions of the edit mode of the interface (Figure 13).

After completing the modeling of roofs for all buildings, we loaded a satellite basemap and the DTM using the BlenderGIS interface. But first, we had to assign the appropriate coordinate projection that matches the one assigned to the created model. To place the created 3D building model on the surface of the DTM, we had to convert it to obj format.

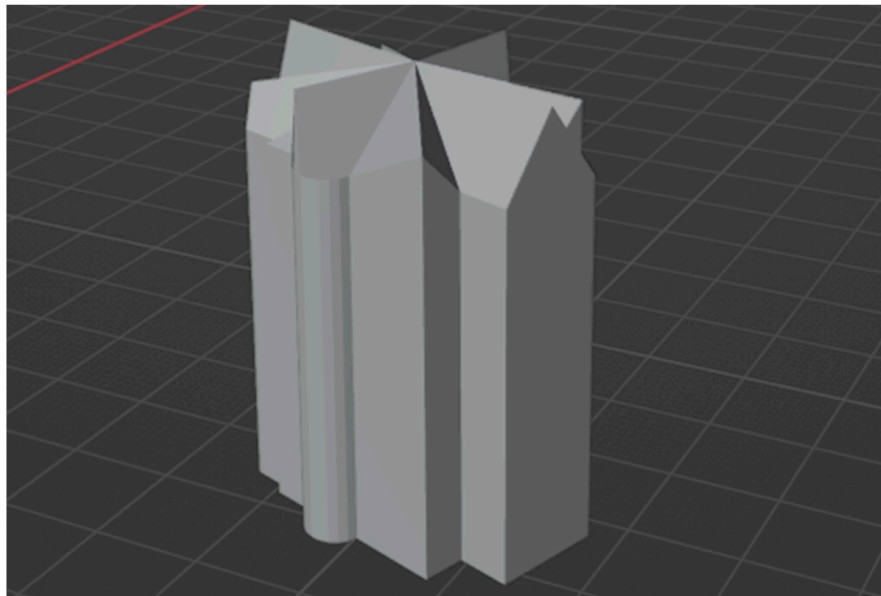

**Figure 13.** Visualization of a sacral building in Blender.

## 4. Results

Using modeling procedures described in the previous chapter, a 3D model of buildings in the wider city center of Ljubljana was created at the LOD2 level of detail (Figure 14). The obtained results were compared with the corresponding section of the three-dimensional representation in Google Earth.

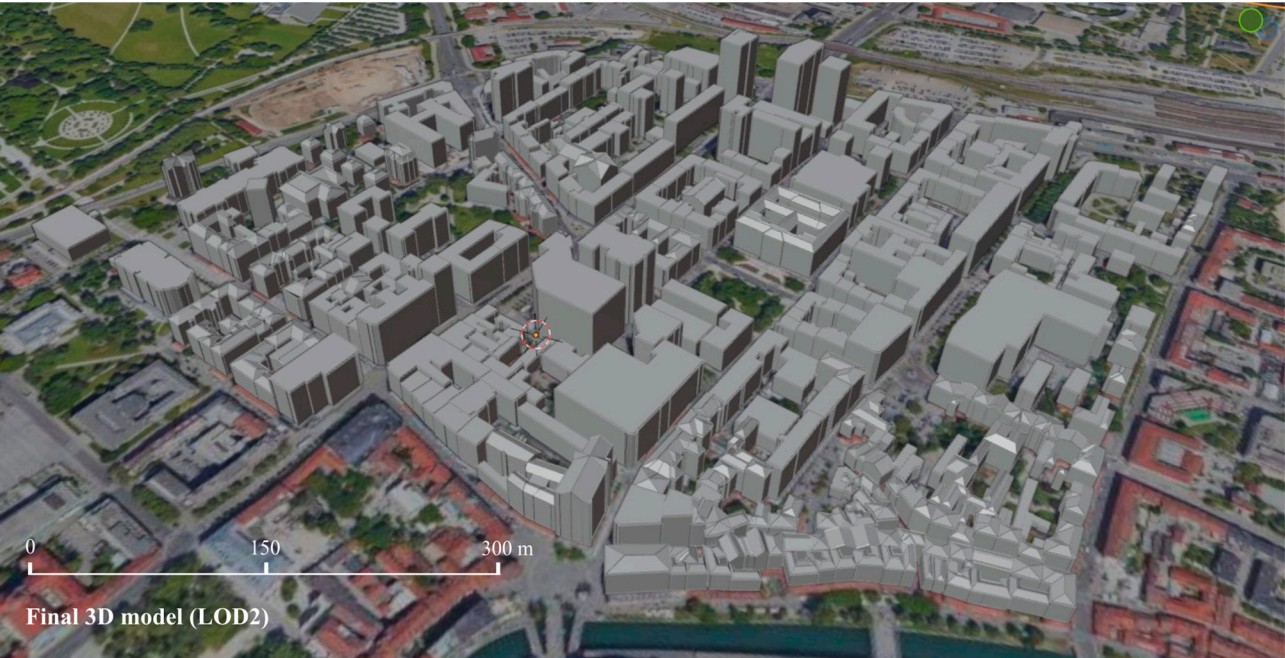

**Figure 14.** The final 3D model in LOD2.

By visual comparison, it was determined that the building models, according to their position, shape, and heights, mostly correspond to the view from Google Earth. However, by inspecting in a more detailed fashion, one can see how certain buildings deviate in height compared to the Google Earth model. Sacred buildings, with their structure and roof skeleton, are greatly simplified compared to the models from Google Earth. This is because bell towers of a churches are significantly higher than the rest of the buildings. Therefore,

they are affected by LiDAR airborne measurements. Hence, the mean value of for the height of this type of building when calculating in the Zonal Statistics tool contributes to an exaggerated mean value of the roof height for the entire building (Figure 15).

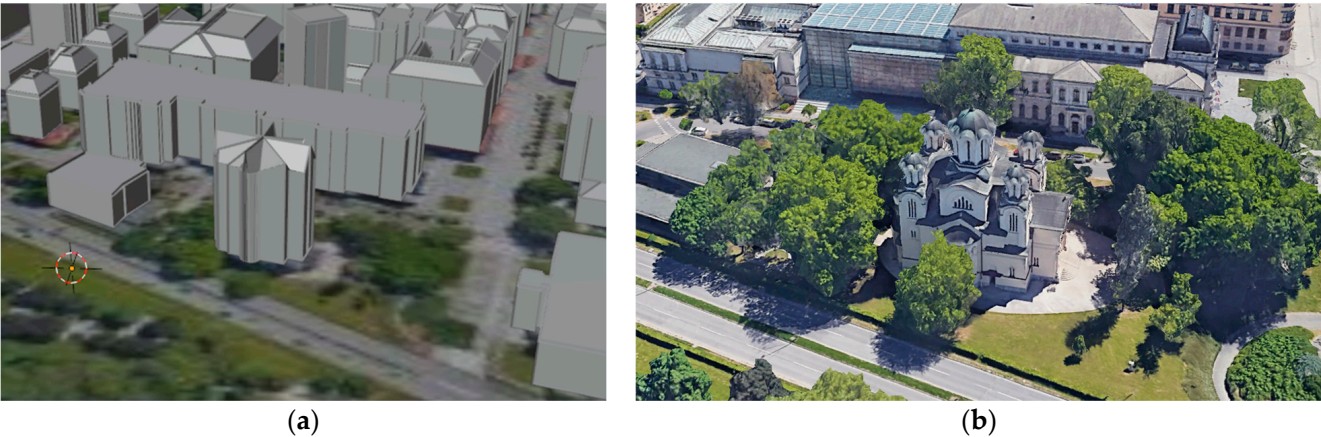

(**a**)          (**b**)

**Figure 15.** Comparison of two representations for a sacred object in: (**a**) Blender; (**b**) Google Earth.

Another comparison was conducted for the business district of Ljubljana using Google Earth (Figure 16). As anticipated, the 3D model closely aligns with the model in Google Earth, particularly in this section of the study area. The reason behind this close visual match user can be attributed to the straightforward and contemporary design of commercial buildings, which feature numerous flat surfaces such as facades and roofs.

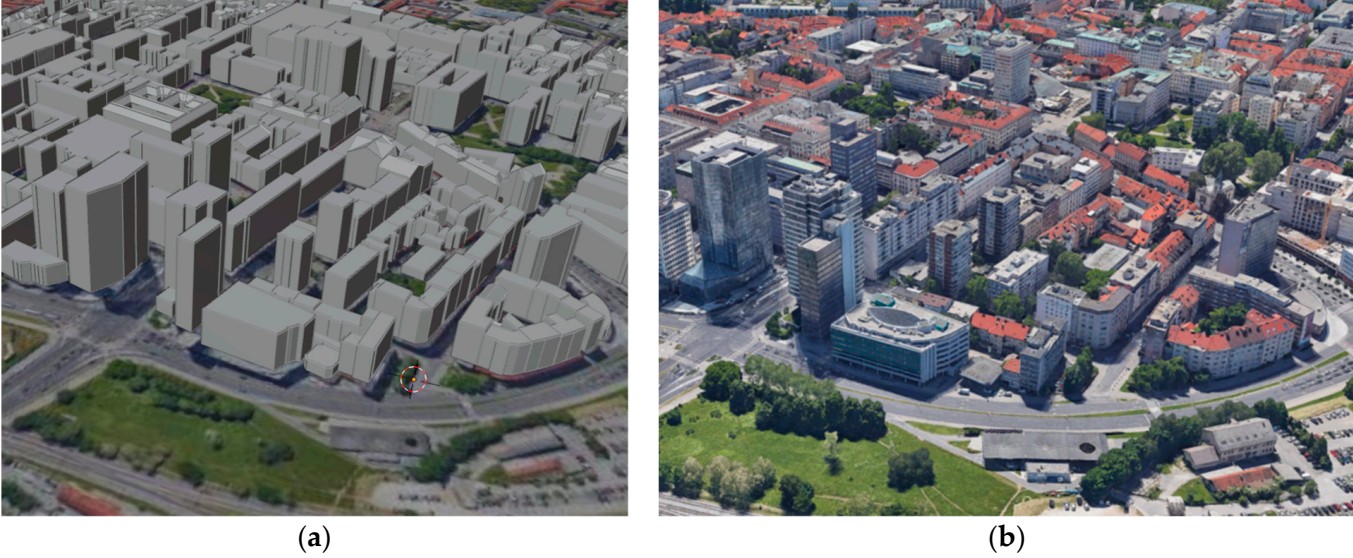

(**a**)          (**b**)

**Figure 16.** View of the business district of Ljubljana, rendered in: (**a**) Blender; (**b**) Google Earth.

Religious buildings are too complex to be automatically modeled. But, that is not the goal of the research at all. It was made as much as possible from the existing data and with the method we used. There are no outlines of the building so one can separate it into multiple segments on the basis of which the religious building would be modeled in more detail in 3D. Tented and pitched roofs are classified in the same category because the modeling process is identical; thus, there is no need to make a difference. Large building behind the religious building is also visually different from the building model in Figure 15b; but, when you understand the research process for which concessions were made and because of the automation of certain elements, such as the height of buildings

as well as the limitation to LOD2, then it is clear that in procedural modeling, one should make certain concessions in the visual sense. This is because the definition of procedural modeling is such that one should not give a faithful representation of the object, but an approximate representation.

## 5. Discussion

While processing spatial data for input into Blender, the handling of OSM data proved to be simple and straightforward, but this data source caused the most inconsistencies in the final model. Availability and spatial coverage of OSM data for the study area is very good as expected. However, when processing OSM data for the attribute table, we noticed that only 82 out of 453 buildings (Figure 17) have one or more attributes describing the external appearance of the building, such as total number of floors, façade color, roof type, or roof material.

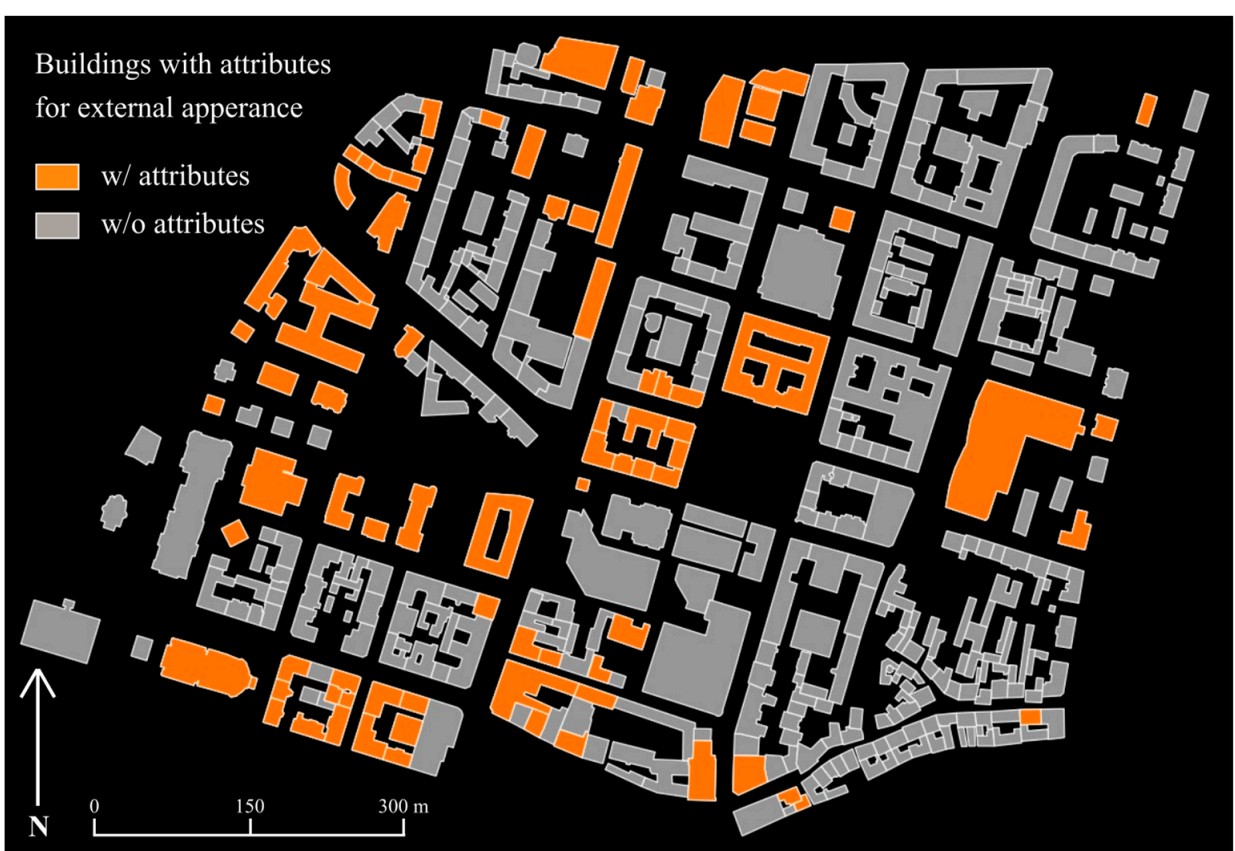

**Figure 17.** Buildings with attributes in OSM describing the external appearance (in orange).

Of the listed 82 buildings that have some of the attributes that describe the appearance of the facade and roofs, only 28 of the 453 buildings contained a roof type (Figure 18). For the remaining buildings, we had to manually enter the roof type in the attribute table.

After completing the data preparation and loading it into Blender, most roof types could be simply generated using one of the three roof construction methods. For 397 roofs, i.e., 87.64% of the total number of roofs, it was not necessary to manually improve the geometry. Such automatically generated roofs are shown in green on the map in Figure 19.

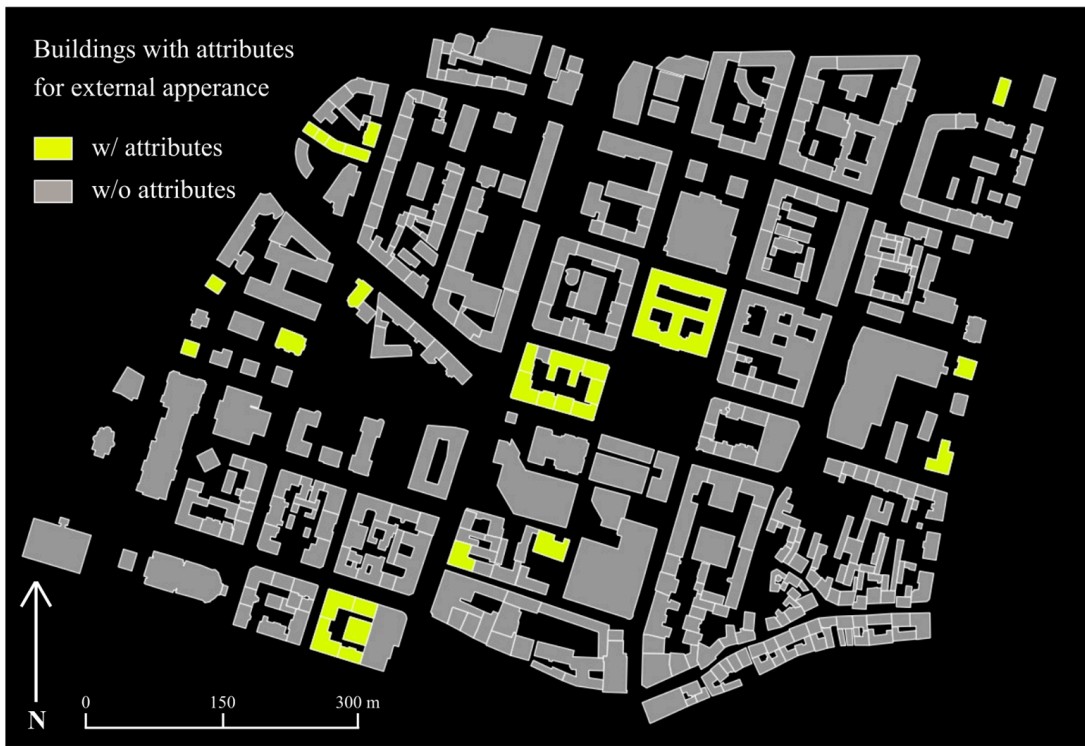

**Figure 18.** Buildings in OSM that have the roof type attribute (in yellow).

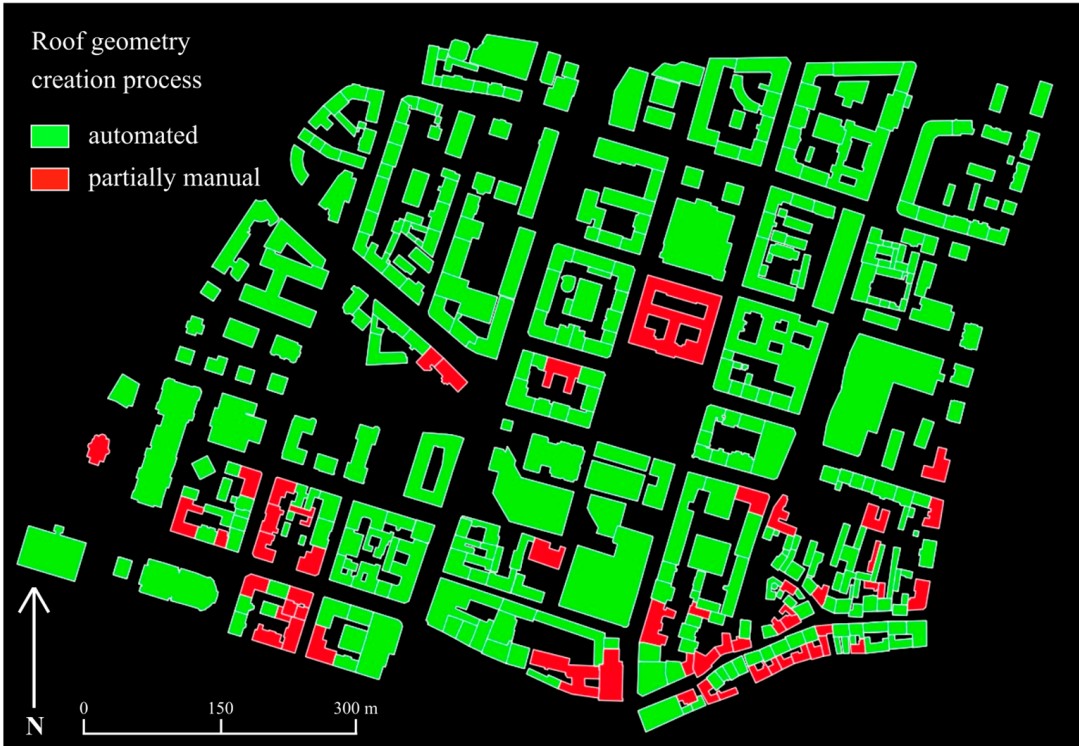

**Figure 19.** Coverage of buildings with automatically generated roof geometry (green) and roof geometries with manual improvement needed (red).

The standard building floor height can vary depending on several factors, including local building codes, architectural design, and the intended use of the building. However, there are common ranges that are often used in building construction. In many countries, a

typical floor height ranges from 2.4 m to 3 m. For the average (or standard) floor height, we decided to use 2.6 m. This value allows for comfortable ceiling heights, while considering structural requirements, mechanical systems, and accommodating various building components (https://cementconcrete.org/building-construction/functional-components-building-structure/3246/, accessed on 19 May 2023). It is important to note that certain types of buildings, such as commercial or office spaces, may have higher ceilings to accommodate additional infrastructure, ventilation, or specialized equipment. In OSM, a total of 78 buildings had the number of floors (building levels in OSM) as a registered attribute. This value was used to compare building heights in OSM and building heights obtained from LiDAR data. The resulting Table A1 is available in Appendix A. For several objects where height differences significantly differ, it was determined that at the time of measurement, construction was undergoing. All buildings with significant differences in height are in the northern part of the study area, which is a business–residential zone.

LiDAR data quality also had an impact in model development, but to a lesser extent than OSM data. Spatial resolution has influence when calculating building heights using the Zonal Statistics tool. The spatial resolution of the point cloud is 5 pt/m$^2$. A higher point density would allow for better filtering of additional contents on the roof such as telecommunication devices and chimneys, since their height should not be part of the statistical calculations. It is presumed that utilizing LIDAR point clouds with higher resolution would lead to a reduced average height of buildings within the surveyed area.

During data processing for Blender, handling of LiDAR and OSM data in our research was mostly manual. First, it was necessary to check the condition of the attribute table and, based on completeness, choose a method to fill in missing data. In this case, we decided to populate the attribute data manually, for as many as 93.82% of the buildings. In the second stage of modeling, 3D models were created in Blender from the preprocessed building outlines and height data. This part of the procedure is fully automated. And last, during the final production phase, i.e., roof modeling using the procedural modeling interface within Blender, the level of automation was 87.64%, while for the rest, it was necessary to edit geometries manually to create a 3D model that better corresponds to reality.

Given that even 93.82% of buildings in the study area did not have data about the roof type, an approach worth considering would be the automation of the classification of roof shapes through machine learning. One of the ways is described in the work of Castagno et al. [48], where automatic recognition of roof types was achieved for a total of 4500 buildings within three different cities. In addition to common roof types, such as tented, pitched, or semi-pitched roofs, the algorithm recognizes flat roofs with air conditioning infrastructure as a special category. LiDAR images, building outlines, and satellite images are used as input data. Satellite images are a good source of information, but due to their two-dimensional nature, they are subject to lower contrast, shadows, and perspective distortion. LiDAR data is used complementary to satellite images due to the possibility of defining the depth and volume of roofs. Building outlines filter data sources by finding parts of LiDAR and satellite imagery that relate to the corresponding building. Building outlines can be extracted using orthophotos or downloaded from OSM as in our case. All three layers must have the same defined coordinate system and be georeferenced, so that they can be combined into a whole. The result is a set of classified roof types with 87% correctly identified roof types. This method would significantly speed up the process of recording roof types for larger areas, and it certainly brings a more accurate classification of roofs compared to visual inspection via Google Earth.

To model pitched roofs, simple mathematically defined operations were used that transform the initial flat surface that follows the edge of the building into a 3D surface. The result of this modeling are simple structures that mostly clearly show the appropriate roof on the ground. For roof models with complex floor plans and a complex skeleton, a better solution would be to use Polyskel, a planar skeleton construction algorithm written in the Python programming language. The Slovak software company Prochitecture has developed the Bypolyskel library with algorithms for the construction of skeletons and surfaces

(https://github.com/prochitecture/bpypolyskel, accessed on 19 May 2023). However, to use this library, whose programming language is also Python, it would be necessary to program your own add-on for Blender, through which the code could be accessed and implemented in the model.

The shortcomings of this research are mainly in the manual editing of OSM data during attribute data processing for the automatic modeling process. It would be optimal to automate the entry of missing data using the abovementioned method of detecting roof types. Furthermore, the achieved level of detail is LOD2, which is not sufficient to store some semantic data. For example, one parameter that is measured, recorded, and modeled in urban analysis is the visibility of certain buildings and parts of the city from certain windows. Since the building models do not have any features on the facade, it would not be possible to store this type of information. It would be necessary to apply machine learning to many images of facades characteristic of the analyzed area; typify them; and then separate them into elementary parts, derive a set of rules according to which they fit into a whole, and then assign such elementary parts to modeled buildings. The level of detail would, thus, be raised to LOD3; and it would also be possible to enter some new semantic data, such as the energy efficiency of the building or the evacuation plan.

The proposed direction of continuing work on 3D building models would be implementation in the CityGML (or CityJSON) standard and assignment of classes and semantic data. Research recommendations point to the use of GIS tools to extract as much information as possible from the LIDAR point cloud, given that such information can serve as a definition for automatically generated 3D models in a later stage of work. It can, therefore, be concluded that the contribution of this work compared to previous research lies in the simple access to LIDAR data and the emphasis on the maximum use of available data within the limits of the possibilities of free open-source software. The described manual steps in the entire procedure also encompass an automatic component in which they are interconnected and executed automatically, as previously described in the research. This is most evident through the programmed sections in Python, which are available at (https://mega.nz/file/sA5S3AiZ#8oZ9_oOIQSlufw9haSR4q69fie5P_gpabWG_OLkrRWE, accessed on 19 May 2023) and (https://mega.nz/file/sZxniIgJ#hm6bytEsegYKmq2Yjg8I7zZdDg76DnySWZhSFuxGYx0, accessed on 19 May 2023).

## 6. Conclusions

Through the integrated usage of freely available LiDAR point cloud and OSM data, we have demonstrated the process of procedural modeling for a large urban area. A crucial aspect of this research lies in data preparation, which aims to automate the modeling process. With LiDAR point clouds, data preparation involves segmenting the point cloud to extract the parts relevant to buildings within our study area, merging different point clouds into one, and filtering out noise and points unrelated to buildings. In the case of OSM data, it is essential to evaluate user-entered attributes, remove unnecessary data, and fill in missing attribute information specific to buildings. These attributes play a critical role in the subsequent stages of 3D modeling. Relevant attributes for building modeling and visualization can be determined and inputted through user intervention or automated machine-learning methods. The level of user intervention in the modeling process may vary depending on the availability and coverage of spatial and semantic source data. For smaller areas, a combination of manual and semi-automatic modeling approaches is acceptable, resulting in a higher level of detail and a more realistic representation. However, for larger areas requiring modeling, procedural modeling techniques [49] are preferable. Besides reducing the need for user intervention and saving time, procedural modeling, with its defined rules and geometric regularities, generates topologically meaningful models that can be further refined and modified.

Point clouds obtained through LiDAR imaging offer a strong foundation for model construction, particularly when combined with topological data and photographs. While established practices and techniques exist for reconstructing detailed building surfaces

from aerial photographs, the widespread implementation of procedural 3D modeling for large-scale areas, encompassing entire cities or national territories, is still limited. Given that LiDAR data obtained through aerial imaging have lower resolution compared to terrestrial or drone LiDAR surveys, the building models produced in this research using a partially automated approach exhibit exceptional accuracy despite the level of detail attained with LiDAR data.

To summarize, procedural modeling is a powerful technique that employs algorithms and rules to generate intricate and realistic digital content. Its capability to create expansive and diverse landscapes and models has contributed to its growing popularity in various fields, such as video game design, film production, architecture, and urban planning. With the continuous advancement of machine-learning techniques, the potential for procedural modeling is boundless, making it an exciting area of research for the future of computer graphics and geovisualization.

**Author Contributions:** Conceptualization, R.N., A.V. and R.Ž.; Methodology, R.N., A.V. and B.P.; Software, R.N. and A.V.; Validation, R.N.; Formal analysis, R.Ž.; Investigation, R.Ž., A.V., R.N. and B.P.; Resources, A.V.; Data curation, B.P.; Writing—original draft, A.V., R.N. and B.P; Writing—review & editing, R.Ž.; Funding acquisition, R.Ž. and R.N. All authors have read and agreed to the published version of the manuscript.

**Funding:** This research received no external funding.

**Data Availability Statement:** Not applicable.

**Conflicts of Interest:** The authors declare no conflict of interest.

## Appendix A

**Table A1.** Height differences between Lidar-derived data and OSM data in meters (m).

| OSM ID | OSM Building Levels | OSM Building Height m | LiDAR Mean Building Height m | Height Difference m | LiDAR Max Building Height m | Height Difference m |
|---|---|---|---|---|---|---|
| 2512227 | 1 | 2.60 | 28.4 | 25.8 | 37.4 | 34.8 |
| 4089569 | 4 | 10.40 | 21.5 | 11.1 | 27.8 | 17.4 |
| 24785631 | 4 | 10.40 | 15.3 | 4.9 | 24.2 | 13.8 |
| 24786527 | 2 | 5.20 | 7.2 | 2.0 | 9.2 | 4.0 |
| 24786528 | 14 | 36.40 | 38.5 | 2.1 | 52.4 | 16.0 |
| 61024039 | 12 | 31.20 | 35.3 | 4.1 | 43.9 | 12.7 |
| 111584761 | 4 | 10.40 | 16.4 | 6.0 | 29.8 | 19.4 |
| 153059527 | 13 | 33.80 | 10.8 | −23.0 | 16.1 | −17.7 |
| 176487482 | 4 | 10.40 | 14.4 | 4.0 | 17.9 | 7.5 |
| 176487486 | 4 | 10.40 | 20.2 | 9.8 | 26.2 | 15.8 |
| 177270343 | 3 | 7.80 | 14.7 | 6.9 | 17.4 | 9.6 |
| 179744268 | 0 | 0.00 | 7.3 | 7.3 | 20.0 | 20.0 |
| 179744269 | 8 | 20.80 | 22.4 | 1.6 | 30.3 | 9.5 |
| 179746610 | 4 | 10.40 | 16.1 | 5.7 | 27.8 | 17.4 |
| 186332693 | 4 | 10.40 | 16.8 | 6.4 | 19.0 | 8.6 |
| 186335765 | 15 | 39.00 | 54.8 | 15.8 | 71.0 | 32.0 |

**Table A1.** *Cont.*

| OSM ID | OSM Building Levels | OSM Building Height m | LiDAR Mean Building Height m | Height Difference m | LiDAR Max Building Height m | Height Difference m |
|---|---|---|---|---|---|---|
| 186335774 | 6 | 15.60 | 24.1 | 8.5 | 28.9 | 13.3 |
| 186513050 | 11 | 28.60 | 32.4 | 3.8 | 37.6 | 9.0 |
| 186513051 | 11 | 28.60 | 34.7 | 6.1 | 37.3 | 8.7 |
| 186515083 | 6 | 15.60 | 16.3 | 0.7 | 33.3 | 17.7 |
| 186518586 | 4 | 10.40 | 19.8 | 9.4 | 23.8 | 13.4 |
| 186518625 | 4 | 10.40 | 19.7 | 9.3 | 24.4 | 14.0 |
| 186547097 | 3 | 7.80 | 15.7 | 7.9 | 21.5 | 13.7 |
| 186547098 | 14 | 36.40 | 41.0 | 4.6 | 44.5 | 8.1 |
| 186547099 | 2 | 5.20 | 14.6 | 9.4 | 33.9 | 28.7 |
| 186547100 | 6 | 15.60 | 19.3 | 3.7 | 24.3 | 8.7 |
| 186547113 | 8 | 20.80 | 24.7 | 3.9 | 29.0 | 8.2 |
| 186547122 | 10 | 26.00 | 24.7 | −1.3 | 40.0 | 14.0 |
| 186547124 | 14 | 36.40 | 41.6 | 5.2 | 45.7 | 9.3 |
| 186547129 | 7 | 18.20 | 27.0 | 8.8 | 31.4 | 13.2 |
| 186547134 | 7 | 18.20 | 22.6 | 4.4 | 27.2 | 9.0 |
| 186547137 | 2 | 5.20 | 9.4 | 4.2 | 13.1 | 7.9 |
| 186549962 | 7 | 18.20 | 19.4 | 1.2 | 23.9 | 5.7 |
| 196794006 | 21 | 54.60 | −12.1 | −66.7 | 81.0 | 26.4 |
| 197017824 | 4 | 10.40 | 22.9 | 12.5 | 26.1 | 15.7 |
| 235168944 | 13 | 33.80 | 49.8 | 16.0 | 60.4 | 26.6 |
| 248803106 | 3 | 7.80 | 16.4 | 8.6 | 21.4 | 13.6 |
| 248803110 | 5 | 13.00 | 3.4 | −9.6 | 12.7 | −0.3 |
| 496313466 | 3 | 7.80 | 15.9 | 8.1 | 20.8 | 13.0 |
| 496313467 | 6 | 15.60 | 22.1 | 6.5 | 27.2 | 11.6 |
| 778984182 | 5 | 13.00 | 19.1 | 6.1 | 24.3 | 11.3 |
| 824372657 | 7 | 18.20 | 23.3 | 5.1 | 28.3 | 10.1 |
| 824372661 | 3 | 7.80 | 12.5 | 4.7 | 17.6 | 9.8 |
| 936397467 | 22 | 57.20 | 4.4 | −52.8 | 81.0 | 23.8 |
| 976077230 | 7 | 18.20 | 0.2 | −18.0 | 27.0 | 8.8 |
| 976077231 | 8 | 20.80 | 0.2 | −20.6 | 30.0 | 9.2 |
| 976077232 | 7 | 18.20 | 0.1 | −18.1 | 7.9 | −10.3 |
| 1030934401 | 2 | 5.20 | 13.6 | 8.4 | 17.8 | 12.6 |
| 1030934402 | 2 | 5.20 | 11.4 | 6.2 | 17.7 | 12.5 |
| 1036836792 | 3 | 7.80 | 14.0 | 6.2 | 19.0 | 11.2 |
| 1040050251 | 4 | 10.40 | 18.1 | 7.7 | 22.7 | 12.3 |
| 1040050252 | 4 | 10.40 | 20.1 | 9.7 | 24.1 | 13.7 |
| 1040050253 | 4 | 10.40 | 20.4 | 10.0 | 23.9 | 13.5 |

**Table A1.** *Cont.*

| | | | | | | |
|---|---|---|---|---|---|---|
| 1040050254 | 4 | 10.40 | 18.7 | 8.3 | 22.2 | 11.8 |
| 1040050255 | 4 | 10.40 | 19.5 | 9.1 | 23.2 | 12.8 |
| 1040050256 | 4 | 10.40 | 19.5 | 9.1 | 23.0 | 12.6 |
| 1040050257 | 4 | 10.40 | 18.0 | 7.6 | 23.5 | 13.1 |
| 1055837765 | 4 | 10.40 | 20.6 | 10.2 | 24.3 | 13.9 |
| 1055837766 | 4 | 10.40 | 20.6 | 10.2 | 23.6 | 13.2 |
| 1055837767 | 4 | 10.40 | 20.5 | 10.1 | 24.7 | 14.3 |
| 1055837768 | 4 | 10.40 | 19.9 | 9.5 | 25.0 | 14.6 |
| 1118759071 | 4 | 10.40 | 15.8 | 5.4 | 18.9 | 8.5 |
| 1118759076 | 4 | 10.40 | 12.3 | 1.9 | 18.7 | 8.3 |
| 1118759078 | 4 | 10.40 | 8.7 | −1.7 | 22.3 | 11.9 |
| 1118759079 | 4 | 10.40 | 19.8 | 9.4 | 22.9 | 12.5 |
| 1118759081 | 4 | 10.40 | 19.1 | 8.7 | 22.8 | 12.4 |
| 1118759082 | 4 | 10.40 | 18.8 | 8.4 | 22.7 | 12.3 |
| 1118759083 | 4 | 10.40 | 14.5 | 4.1 | 23.4 | 13.0 |
| 1118759084 | 4 | 10.40 | 12.0 | 1.6 | 19.6 | 9.2 |
| 1118759085 | 4 | 10.40 | 13.5 | 3.1 | 21.6 | 11.2 |
| 1120198927 | 4 | 10.40 | 20.3 | 9.9 | 24.7 | 14.3 |
| 1120210047 | 7 | 18.20 | 20.8 | 2.6 | 23.7 | 5.5 |
| 1120210048 | 7 | 18.20 | 23.8 | 5.6 | 25.7 | 7.5 |
| 1120210049 | 7 | 18.20 | 22.2 | 4.0 | 24.4 | 6.2 |
| 1121513868 | 3 | 7.80 | 14.1 | 6.3 | 17.3 | 9.5 |
| 186547113 | 8 | 20.80 | 23.4 | 2.6 | 34.6 | 13.8 |
| 153059527 | 13 | 33.80 | 41.6 | 7.8 | 70.6 | 36.8 |
| 153059527 | 13 | 33.80 | 8.0 | −25.8 | 12.4 | −21.4 |

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
