# Peer review of "Automatic 3D Building Model Generation from Airborne LiDAR Data and OpenStreetMap Using Procedural Modeling"

_information, doi:10.3390/info14070394_

Round 1
Reviewer 1 Report
See https://www.geopipe.ai/ and https://www.nvidia.com/en-us/omniverse/ and MAYBE CityEngine from Esri. Here's my question -- how much do these commercial software packages overlap with this work? If you want to do an even more comprehensive analysis, see Mapbox and Cesium, at least to note their existence.
Is there an extra line? (see #556)
This critique is more about my preference, but I like to encourage writers to consider doing the following:
* change any "in order to" to just "to"
* you have 15 instances of "it is" -- try to rewrite each by asking, "what makes that process necessary?" or "who made that decision?" -- I try to remove passive tense when possible
* try to change "can be" to who does the "can" -- for example, "something can be downloaded" changes to "Users can download"
* want to improve even more? Look for phrases like line 99 -- change to "Researchers emphasized..." Some people disagree with my style because you get changes like line 101 ... it becomes "...the algorithm detects" ... personally, I still think that reads more smoothly and removes needless passive tense. Dig around -- I think you can enjoy rewriting all this passive wording academic writing seems to espouse.
* see the paragraph starting with 49 -- please change long sentences into bulleted lists
Author Response
Information MDPI
Manuscript title: Automatic 3D Building Model Generation from Airborne Li-DAR Data and OpenStreetMap using Procedural Modeling
Answers to the comments and suggestions:
Dear Editors and Reviewers,
Thank you very much for provision of remarks and suggestions. All are accepted and corrected in text. The authors hope that corrections and further explanations which are provided according to your comments and suggestions improved quality of the manuscript up to the satisfactory level it can be accepted for publication in Information. Please see all author responds in word file.

Reviewer 2 Report
The paper entitled "Automatic 3D Building Model Generation from Airborne Li-DAR Data and OpenStreetMap using Procedural Modeling" presents a study about the generation of 3D models of buildings for a large-scale urban area starting from available airborne LiDAR data and spatial data from openly available databases. One of the main points is the reconstruction of building profiles according to a Level of Detail equals to 2, that, according to standards is "outline of the building/structure represented as a solid object with principal architectural features included using generic components".
In this regard, authors recognize four types of roofing systems: flat, tented or pitched, and gable. Then, tented and pitched roofs are classified in the same category. My concern is about the possibility to have clustered buildings with different heights. For instance, inspecting Figure 16 a and b, it is possible to see that the building on the left of the religious building is reconstructed with on height of elevation in (a) whether architectural features suggest two different clustered buildings with two different heights.
Similar comments about the large building behind the religious building. Could the authors add some further comments on that point?
Please check the English language. In particular correct Mai 2023 -> May 2023 in the whole manuscript.
Author Response

(The authors gave the same response as above.)

Reviewer 3 Report
Dear authors,
Thank you very much for your research. I hope my reviews can be helpful to improve your work. Detailed comments are included in the attached PDF.
Thank you.

Overall, the manuscript is well-written though there are minor errors in English.
Author Response

(The authors gave the same response as above.)

Round 2
Reviewer 3 Report
Dear Authors,
Thank you for your revised manuscript, information-2463034. I have some minor recommendations for your work. I hope the comments included in the attached PDF document help.
Thank you.

The manuscript needs a few minor corrections in English. Please refer to the attached PDF document for detail.
Author Response
Dear reviewer 3 and editors
Thank you for all suggestions. We accepted all and corrected manuscript, so we are sending back a new version. Please let us know if anything else is needed.
Here are reviewer findings and answers:
Reviewer: Please include description of the color scheme of Figure 2(a). It is not clear what red, yellow, and green colors mean in the figure.
Authors: Corrected
Reviewer: Please provide the legend, scale bar, and north arrow for the map. Again, the labels are too small to read. This map has a different color scheme and scale from the map above.
Authors: This map was deleted. We left only in Track changes but now we can send clean version so there is no confusion.
Reviewer: Please check with the Journal policy if data source for the map should be cited in the caption.
Authors: Accepted. The copyright was cited in the caption.
at figure 4.
Reviewer: Please include the legend that provides information of the color scheme in Figure 4. It is not clear what brown, green, orange, and black colors mean.
Authors: Corrected
Reviewer: It should be written as "$125" here. If you want to mention the price, you may want to add the year information, too.
Authors: Corrected
Reviewer: Redundant words: "files" can be removed.
Authors: Removed
Reviewer: Please provide the citation for each software tool introduced.
Authors: Corrected
Reviewer: Please check with the Journal policy for citing a URL in the manuscript.
Authors: Checked, and can be found at published articles, e.g. https://www.mdpi.com/2078-2489/11/2/125/pdf?version=1582551862
Reviewer: Please check with the Journal policy for citing a URL in the manuscript.
Authors: Same as previous
Now we are sending clean version, and if needed we can send with Track changes on.
Best regards,
authors
